

# Extratropical Age of Air trends and causative factors in climate projection simulations

Petr Šácha[1,2], Roland Eichinger[4,3], Hella Garny[3,4], Petr Pišoft[2], Simone Dietmüller[3], Laura de la Torre[1], David A. Plummer [5], Patrick Jöckel [3], Olaf Morgenstern[6], Guang Zeng[6], Neal Butchart[7] and Juan A. Añel[1]

[1]EPhysLab, Faculty of Sciences, Universidade de Vigo, Ourense, Spain.
[2]Charles University in Prague, Faculty of Mathematics and Physics, Department of Atmospheric Physics, Prague, Czech Republic.
[3]Deutsches Zentrum für Luft- und Raumfahrt (DLR), Institut für Physik der Atmosphäre, Germany.
[4]Ludwig Maximilians Universität, Institut für Meteorologie, Munich, Germany.
[5]Environment and Climate Change Canada, Climate Research Division, Montréal, QC, Canada.
[6]National Institute of Water and Atmospheric Research (NIWA), Wellington, New Zealand.
[7]Met Office Hadley Centre, Exeter, UK.

*Correspondence to*: Petr Šácha (sacha@uvigo.es)

**Abstract.** Climate model simulations show a Brewer-Dobson circulation (BDC) acceleration in the course of climate change. While the mechanisms for the BDC strengthening are well understood, there are still open questions concerning its dynamical driving. Mean age of stratospheric air (AoA) is a useful transport diagnostic for accessing changes of the BDC. Analysing AoA from a subset of Chemistry Climate Model Initiative part 1 climate projection simulations, we find a remarkable agreement between most of the models in simulating the largest negative AoA trends in the extratropical lower to middle stratosphere of both hemispheres (approximately between 20 gpkm and 25 gpkm and 20°- 50°N/S). We show that the occurrence of AoA trend minima in those regions is directly related to the climatological AoA distribution being sensitive to an upward shift of the circulation in response to a climate change. But also other factors like a reduction of aging by mixing (AbM) and residual circulation transit times (RCTTs) contribute to the AoA distribution changes by widening the AoA isolines. Furthermore we analyze the time evolution of AbM and RCTT trends in the extratropics and examine the connection to possible drivers like local residual circulation strength, net tropical upwelling and wave driving. However, after the correction for a vertical shift of pressure levels, we find only seasonally significant trends of residual circulation strength and zonal mean wave forcing (resolved and unresolved) without a clear relation between the trends of the analyzed quantities. This indicates that additional causative factors may influence the AoA, RCTT and AbM trends. In this study, we postulate that the shrinkage of the stratosphere has the potential to influence the RCTT and AbM trends and thereby cause additional AoA changes over time.



# 1 Introduction

A global mass meridional circulation in the stratosphere, the Brewer-Dobson circulation (BDC), has been discovered by Brewer (1949) and by Dobson (1956) through analysis of trace gas distributions. Climate model simulations robustly show

that the BDC accelerates in connection to the greenhouse gas induced climate change (Shepherd and McLandress, 2011; Palmeiro et al., 2014) and this acceleration dominates the stratospheric changes in climate model projections (Butchart, 2014). Recently, Polvani et al. (2017, 2018) and Morgenstern et al. (2018) showed that ozone depleting substances are key drivers of BDC trends with the potential to considerably reduce the trends in the future. However, the physical cause behind the BDC changes, in particular the role of various sources of wave driving for the circulation and its variations (Cohen et al.,

2014), remains an open issue.

The BDC consists of two separate branches - a 'shallow branch' in the subtropical lower stratosphere (LS) and a 'deep branch' higher in the middle atmosphere (Andrews et al., 1987; Plumb, 2002; Birner and Boenisch, 2011). Both BDC branches are currently considered to be driven primarily by resolved waves of different scales (Plumb, 2002) plus contributions from gravity waves (GWs) in the upper stratosphere and mesosphere (Andrews et al., 1987) as well as above

the subtropical jet (McLandress and Shepherd, 2009). Li et al. (2008), Okamoto et al. (2011) and Butchart (2014) highlighted the role of GW drag (GWD) and especially of orographic GWD (OGWD) changes for driving the trend of the shallow BDC branch. But, there are also indications that changes in the unresolved wave drag are often compensated by changes in the resolved wave driving (Mclandress and McFarlane, 1993; Cohen et al., 2013; Cohen et al., 2014; Sigmond and Shepherd, 2014), which hardly makes it possible to clearly separate the two effects (e.g. by the Downward Control (DC)

principle (Haynes et al., 1991)).

In this study, we analyze the mean age of stratospheric air (AoA; Hall and Plumb, 1994) trends and their causative factors in the REF-C2 scenario (see Eyring et al., 2013) from a subset of models participating in the Chemistry Climate Model Initiative part 1 (CCMI-1; Morgenstern et al., 2017). AoA is a useful transport diagnostic and one of the best tools for accessing the BDC change (Butchart, 2014). For different methods of definition and chemistry climate model inter-

comparison of an age-of-air in the troposphere refer to Krol et al. (2018). In the first part of our paper, we highlight a remarkable agreement between the majority of models in projecting the strongest negative AoA trends (global or local minimum) in the extratropical lower to middle stratosphere of both hemispheres. The extratropical regions of strongest AoA trend have previously been noted by e.g. Okamoto et al. (2011), Li et al. (2012) and Butchart (2014). Studying the full AoA spectrum, Li et al. (2012) have attributed the existence of strong AoA trends in the extratropics to the effect of a

strengthening residual circulation and a weakening of the so-called "recirculation" (in-mixing of old stratospheric air into the tropical pipe).

In following sections of results, where we produce a more in-depth analysis of the kinematic and dynamic changes corresponding with the regions of strongest AoA trends, we use the Canadian Middle Atmosphere Model (CMAM; Scinocca et al., 2008) simulation. CMAM uses relatively advanced orographic (Scinocca et al., 2000) and non-orographic (Scinocca,





2003) GW parameterization schemes and have been chosen, as there are previous studies on the topic of the BDC (McLandress and Shepherd, 2009) and wave driving (Shepherd and McLandress, 2011) response to climate change that have been based on CMAM. Also, the issue of compensation between resolved and unresolved wave driving has been studied extensively for CMAM (Sigmond and Shepherd, 2014).

First, we illustrate that the minimal AoA trends in the extratropical lower to middle stratosphere in CMAM are connected

with the climatological AoA distribution. In this region, the AoA distribution is sensitive to the vertical shift of the pressure levels under climate change (Lübken et al., 2009) as well as to the widening of the AoA isolines (see the scheme in Fig.1). The isoline widening (not to be confused with the circulation widening) is due to a combination of the upward shift itself and a decrease in AoA. A decomposition of AoA into residual circulation transit times (RCTTs; Birner and Boenisch, 2011) and aging by mixing (AbM; Garny et al., 2014; Ploeger et al., 2015a) shows us additional contribution to the AoA isoline

widening by the AbM reduction.

In the final section of results, we investigate possible causative factors of AbM and RCTT trends with a focus on the hypothesis of a strengthening residual mean circulation in the shallow BDC branch (e.g. Li et al., 2012; Garny et al., 2014; Ploeger et al., 2015a) driven by changes in resolved and unresolved extratropical wave forcing (e.g. Okamoto et al., 2011; Shepherd and McLandress, 2011; Butchart, 2014). However, after the correction to the vertical shift of pressure layers, a

clear connection between the acceleration of the residual circulation, stronger wave driving in the extratropical lower to middle stratosphere, net tropical upwelling and the time evolution of the RCTT and especially AbM trends is not found. On this basis we argue that additional mechanisms may be acting. Namely, in the discussion section, we formulate a hypothesis about a possible impact of a variable shift of pressure levels in the stratosphere under climate change (stratospheric shrinkage; Lübken et al., 2009; Berger and Lübken, 2011) for the AoA (RCTT and AbM) trends.

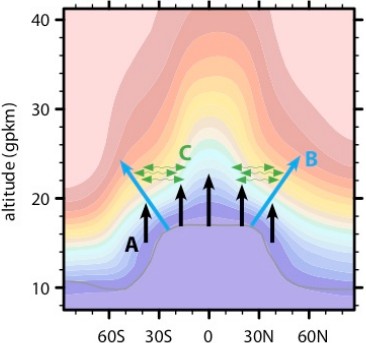


**Figure 1.** Schematic illustration of the location and direction of the effects of the upward shift trend (A), the maximal AoA gradient (B) and the aging by mixing decrease (C) on the AoA trend. The colors indicate the climatological zonal mean AoA distribution of the 1960-2000 period in the CMAM REF-C2 simulation.





## 2 Methodology

### 2.1 Data

Our methodology is motivated with the intention to diagnose the effect of the vertical shift of the circulation due to tropospheric warming and stratospheric cooling (Shepherd and McLandress, 2011; Singh and O'Gorman, 2012; Oberländer-Hayn et al., 2016). For this goal, we have chosen to base our analysis on interpolation to the geopotential height vertical coordinate as an equivalent to the geometric height (for details on the difference between geometric and geopotential height, which is variable with altitude, refer to Andrews et al., 1987). The method of interpolation and vertical shift analysis is described in Section 2.2. Among the models that participate in the CCMI-1 project, we were able to apply this methodology to monthly mean data of five chemistry-climate models (CCMs) – 1) CMAM, 2) the Goddard Earth Observing System CCM (GEOSCCM, Pawson et al., 2008), 3) the ECHAM/MESSy Atmospheric Chemistry model (EMAC; Jöckel et al., 2016) in two setups with different vertical resolution L47 (r2i1p1 ensemble member) and L90, 4) HadGEM3-ES (Hardiman et al., 2017) and 5) NIWA-UKCA (Morgenstern et al., 2009) ensemble of REF-C2 simulations. This selection of models is based on the availability of the required variables for our analysis and applicability of the method described in Section 2.2.

Following Dietmüller et al. (2018), the AoA data require additional modification. For each time step, the AoA value at the tropical tropopause (between 10°S and 10°N) is subtracted from the AoA values everywhere in the stratosphere (so that AoA=0 at the tropopause). This ensures consistency between the simulations and filters the effect of an increasing path the air has to travel before entering the stratosphere due to the tropopause rise over the period of the study (Held, 1982; Santer et al., 2003, Añel et al., 2006, Shepherd and McLandress, 2011, Oberländer-Hayn et al., 2016, Abalos et al., 2017).

The studied period 1960-2100 has been divided into three parts in agreement with common periods of the REF-C2 model outputs: 1960-2000 (regarded as reference period in our study; Ref), 2000-2050 (near future; NF) and 2050-2100 (future, F). Those periods correspond well with the ozone depletion and projected recovery (Dhomse et al., 2018; and Fig. 1S in the Supplement for the ozone evolution in CMAM REF-C2 simulation), which has been highlighted to play a crucial role for driving the BDC trends (Polvani et al., 2017, 2018; Morgenstern et al., 2018).

In Section 3.2, we analyze the mechanisms of the occurence of the minimal AoA trends in the extratropical stratosphere and highlight the important role of decreasing RCTT and AbM for the AoA isoline widening. RCTTs are calculated according to the method of Birner and Bönisch (2011) based on residual circulation backward trajectories. AbM is estimated as the difference between AoA and RCTTs (see Garny et al. 2014, Dietmüller et al. 2017, 2018). This means that AbM includes all sorts of resolved and unresolved mixing (see Dietmüller et al., 2017). Due to the methodology of their computation (initialization of backward trajectories), RCTTs and therefore also AbM data are available starting from the year 1970.

For analysis of the causative factors of AbM and RCTT trends in Section 3.3, we use CMAM REF-C2 monthly OGWD, non-orographic gravity wave drag (NOGWD), EPFD and residual mean velocities ($\overline{v}^*$, $\overline{w}^*$). These quantities are also interpolated to the geopotential height vertical coordinate by means of collocated monthly geopotential data. We decided not to distinguish between the OGWD and NOGWD in the analysis, but take the sum of the two (GWD), partly because we have



zonal mean monthly averaged data only and partly because NOGWD parametrization schemes are usually tuned to have only little influence on the LS (Scinocca, 2003).

Throughout the paper, we use information about the tropopause and turn-around latitudes as a measure for BDC widening (Hardiman et al., 2014). The tropopause is computed as a lapse rate first tropopause using the WMO (1957) definition. The turn-around latitudes are computed as the first latitude with a monthly mean vertical residual velocity being lower or equal to zero going poleward from the equator on the respective hemisphere and geopotential level.

Trends of all variables have been estimated by the Theil-Sen estimator (Theil, 1950; Sen, 1968) and their significance have been computed using the Mann-Kendall test (Mann, 1945; Kendall, 1975). Where applicable, the statistical significance of differences and correlations has been computed by a Student's t-test.

## 2.2 Method

The monthly mean data of all analysed quantities (in the form *Y(time, plvl, lat, lon)*, where *Y* is a scalar function, *time* is a time step, *plvl* pressure or hybrid level depending on a model, *lat* is latitude and *lon* stands for longitude) have been linearly interpolated from model levels to the equidistant geopotential height vertical coordinate ($0 < 1.25 < ... < 70$ gpkm) using collocated geopotential height values $\phi(time, plvl, lat, lon)$ normalized by the standard gravity at the mean sea level (9.80665 $ms^{-2}$). The transformation of a vertical coordinate has a direct influence on the value of trends, which can be illustrated by approximating the trend by a local time derivative and by a simple application of the chain rule. Having two scalar functions *Y(time, plvl, lat, lon)* and *Y'(time, $\phi(time, plvl, lat, lon)$, lat, lon)*, their local time derivatives in pressure and geopotential coordinates are related as follows:

$$\frac{\partial Y}{\partial t} = \frac{\partial Y'}{\partial t} + \frac{\partial Y'}{\partial \phi} \frac{\partial \phi}{\partial t}, \qquad (1)$$

where $\partial$ denotes a partial derivative and *t* is time. In meteorology we generally do not distinguish between *Y* and *Y'*. For AoA (*Y=Y'=AoA*) eq. (1) yields:

$$\left.\frac{\partial AoA}{\partial t}\right|_p = \left.\frac{\partial AoA}{\partial t}\right|_\phi + \frac{\partial AoA}{\partial \phi} \frac{\partial \phi}{\partial t}. \qquad (2)$$

Here the local derivative of AoA in geopotential height vertical coordinate is denoted by subscript $\phi$ and in pressure coordinates by subscript *p*.

In Section 3.2, we estimate and subtract the effect of the vertical shift of pressure levels on the computation of trends (the second term in eqs. (1) and (2)). Later in the text we call this procedure a correction to the vertical shift of the pressure levels. The correction is based on modification of the geopotential height field (to which we interpolate) so that it does not have a trend (in the long term sense $\frac{\partial \phi}{\partial t}$ is zero) and the second term in eqs. (1) and (2)) vanishes. The correction is





implemented as follows: Where the pressure levels have a significant vertical shift trend, its monthly value (decadal trend divided by the number of months, 120) is cumulatively subtracted from the geopotential field used to interpolate AoA (and other quantities). This is demonstrated in eq. (3) for the correction in the F period:

$$\phi^c(i,p,\text{lat}) = \phi(i,p,\text{lat}) - t^{\text{ref}}(p,\text{lat}) \cdot n^{\text{ref}} - t^{\text{nf}}(p,\text{lat}) \cdot n^{\text{nf}} - t^f(p,\text{lat}) \cdot i_{i=0,..,n^f}. \tag{3}$$

$\phi^c$ is the geopotential height after correction that corresponds to a pressure level $p$ and meridional position *lat* in the month $i$

of the F period, $t$ is the pressure level trend (gpm month$^{-1}$) corresponding to $p$ and *lat* in the given period, and $n$ is the number of months in the respective period (Ref, NF, F). Except for interannual and interseasonal variations, the $\phi^c$ vertical coordinate corresponds with the geopotential height of pressure levels at the starting years of the analysis. The resulting trends corrected to the vertical shift of pressure levels are denoted by a superscript [c] further in the text.

The interpolation to the geopotential height vertical coordinate has been performed for monthly mean data (AoA, RCTT) or

for zonal mean monthly mean fields (Eliassen-Palm flux divergence (EPFD), GWD, residual mean velocities). From theory, the interpolation should be made on the finest scale (spatiotemporal) possible. We tried to estimate an upper boundary of the interpolation-connected error by considering an extreme case - we confronted the zonal mean AoA climatology in the 1960-2000 period computed from 1) daily 3D AoA data interpolated on daily basis and 2) from climatological zonal mean AoA data interpolated using the 1960-2000 mean zonal mean geopotential data. This is an extreme case, because in our analysis

we are interpolating at least monthly mean zonal averages. The difference of resulting AoA climatologies is at maximum around ±0.05 year in the polar regions (not shown). Also in other regions (and in the extratropical stratosphere) the upper estimate of the error connected with interpolation reaches at maximum 1% of the AoA climatological value. This shows that the application of our methodology to the monthly mean zonal averages will not qualitatively affect our results, especially in the extratropical stratosphere.

**2.3 Additional diagnostics**

Note that, in the process of changing to a different coordinate system, only values of the original quantities are interpolated. This may be confusing esp. in the case of residual mean vertical velocity ($\overline{w^*}$), which is usually computed in the models as *Pa s$^{-1}$* as most models use hybrid-pressure coordinates (including CMAM; see Table 3 in Morgenstern et al., 2017) and has to be transformed to *m s$^{-1}$* for the standard output. As described in the Supplement of Dietmüller et al. (2018), there are some

inconsistencies between the CCMI-1 models regarding this transformation. It was suggested by the CCMI-1 data request to use the log-pressure relationship for this transformation with a scale height of H = 6950 m. In our study we interpolate $\overline{w^*}$ (log-pressure, given on pressure levels) to the geopotential height vertical coordinate. Also, the procedure of correction for the vertical shift of pressure levels affects only the distribution of $\overline{w^*}$ (log-pressure) in the modified geopotential coordinate, where we compute the trends, not the units of $\overline{w^*}$ (log-pressure) themselves. Later in the text, we are stating several times

that the correction influences the process of the trend computation only, and that it cannot account e.g. for the influence of shrinkage on the vertical velocity in log-pressure coordinates (time dependence of the relationship between log-pressure and





geopotential meter). The validity of the assumption of a constant scale height, which does not take into account the variable vertical shift of pressure levels, is discussed in Section 4.2.

In Sections 3.2 and 3.3 we analyze net tropical upwelling trends and trends of spatially averaged local residual circulation and wave driving. Those quantities in form of mass fluxes or forces are computed from the original pressure (log-pressure) data interpolated to the geopotential height vertical coordinate. Unlike in pressure, in the geopotential height vertical coordinate system, mass flux (force) has to be computed as a product of velocity (acceleration) and density, which is not a standard output in the CCMI-1 REFC2 simulations. In our analysis, density is computed using the state equation for dry air.

However, the net upwelling mass flux trend is dominated by the density trends (negative trends after the correction for the vertical shift, see Fig. 2S and Tab. 3S). Therefore we define a kinematic proxy for the mass flux in the form

$$\left\langle \overline{\rho(z,lat)} \right\rangle_{period} \overline{w^*(z,lat)},$$ where overbar denotes the zonal mean and $\left\langle \ \right\rangle_{period}$ denotes the average across a period.

The net tropical upwelling kinematical proxy (UP) is then computed by three different methods: 1) direct integration of the mass flux proxy between the turn-around latitudes (UP$_{wstar}$), 2) evaluation of the residual mean stream function with a

vertical integral of the mass flux proxy connected with $\overline{v^*}$ at the turn-around latitudes between 20 and 40 gpkm (UP$_{vstar}$) and 3) usage of the quasi-geostrophic version of the DC integral at the turn-around latitudes between 20 and 40 gpkm with a net

force in the form $\left\langle \overline{\rho(z,lat)} \right\rangle_{period} (\overline{EPFD(z,lat)} + \overline{GWD(z,lat)})$ (UP$_{DC}$). See e.g. Okamoto et al. (2011), Abalos et al.

(2015) for more information on the different methods for tropical upwelling computation. The meridional and vertical integration was performed using classical Simpson's rule (Süli and Mayers, 2003).

The average density is also used to define the local residual circulation strength (RC) in the form

$$\left\langle \overline{\rho(z,lat)} \right\rangle_{period} \sqrt{\overline{v^*}^2 + \overline{w^*}^2}.$$ Neglecting the density trend allows us to focus on the acceleration of the circulation (which

is linked with kinematic variables like AoA, RCTT) while still being able to weight the contributions of different vertical levels to the spatial averages from which we compute the trends in Sections 3.2 and 3.3.

## 3 Results

### 3.1 Extratropical AoA trends in the model simulations

In Fig. 2, the trends of zonal mean AoA for the subset of CCMI-1 simulations are shown for the three periods (Ref, NF and F). The contour lines display the climatological AoA distribution of the respective period. In all periods we see that the analysed CCMI-1 REF-C2 simulations show the maximum AoA gradient in the region between the tropical LS and the extratropcial lower to middle stratosphere (illustrated in the scheme in Fig. 1).





Here, we focus on the inter-model agreement in projecting the largest negative trend (global or local minimum) in the extratropical lower to middle stratosphere of both hemispheres. The location of those minima do slightly vary between the models, but can be found in most cases approximately between 20 gpkm and 25 gpkm and 20°- 50°N (ExNH) and 20°- 50°S (ExSH). As long as we do not discuss hemispheric differences, those regions are referred to as Ex regions in the following text. Presence of global or local extremes of the AoA changes in the extratropics also in other CCMI-1 simulations can be

seen in Figs. 1 and 3 in Eichinger et al. (2018).

The best agreement in projecting the minimal trend in the Ex regions is in the NF and F period (Fig. 2). In the NF period, the AoA trends from all analyzed simulations display a well-pronounced, localized NH minimum (on the analyzed vertical domain) in the ExNH region. In the NF period, the ExSH AoA trend minima are only local extremes in NIWA and CMAM and have a different structure in GEOS and EMAC-L47. In the F period, there is a pronounced, localized AoA trend

minimum in all simulations except EMAC-L47 in the ExNH region and EMAC-L47 and HADGEM in the ExSH region. Particularly in the EMAC-L47 simulation, the trend is strongest in the polar regions below/above 30 gpkm in the NH/SH. In the Ref period, the models agree only in projecting strong negative AoA trends in the ExSH region. In the NH, the trends are small or more wide-spread in a broader region with the minimum at the pole.

A localized minimum of the AoA trend in the ExSH region for the 1965-2000 period and in both Ex regions for the 2000-

2080 period is visible also in Fig. 3 in Polvani et al. (2018) for their "All-forcings" simulation. In their study, Polvani et al. (2018) apply the Whole Atmosphere Community Climate Model (Marsh et al., 2013; Solomon et al., 2015; Garcia et al., 2017), which is forced as per the CCMI-1 specifications of scenario REF-C2. The localized and almost symmetric AoA trend features in the Ex regions (best pronounced in our analysis in CMAM, HADGEM and NIWA) are collocated with the maximum climatological AoA gradient. This suggests that the trend minima are a geometric consequence of the

climatological AoA distribution and its future changes that are aligned with the direction of the gradient. We investigate this in the next subsection and propose possible causes for the changes of the AoA distribution.




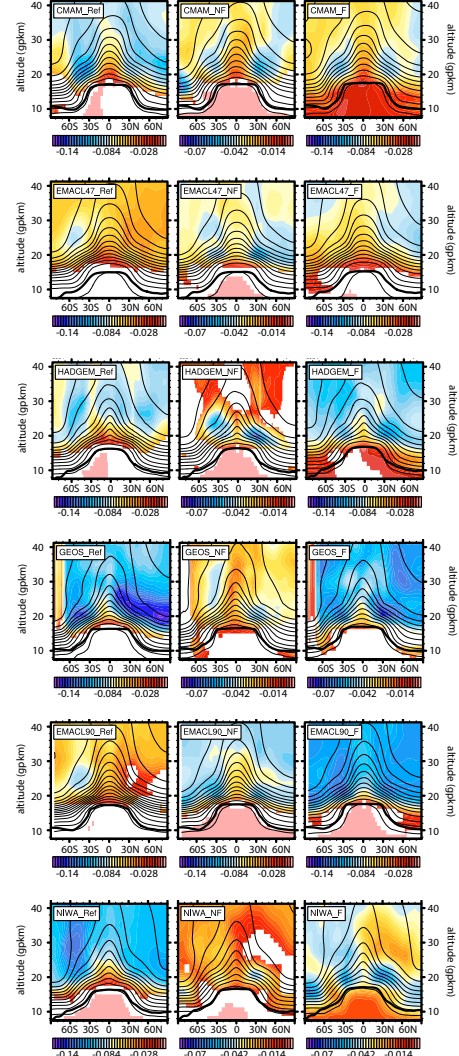

**Figure 2.** Zonal mean AoA trends [year/decade] (colors) and AoA climatology (contours) of the analysed CCMI-1-REFC2 simulations. The left column shows the Ref period, the middle column the NF period and the right column the F period. The vertical axis is in geopotential kilometers [gpkm]. The color bar differs between the periods (except EMAC-L90) to catch the



particular trend structure best. The mean tropopause position is indicated by the bold black line. White regions mark where the significance level of the trends does not exceed 95%.

The time evolution of the AoA trends is in agreement with the effect of the phasing out of the ozone depleting substances, which will lead to a reduction of the BDC trends in future decades (Polvani et al. 2017, 2018; Morgenstern et al., 2018). In Fig. 2, except for the EMAC-L90 REF-C2 simulation, all analyzed simulations are in agreement with this (note that the color bar is different between the periods). The strongest negative AoA trends (globally as well as in the Ex regions) can be detected in the Ref period and thereafter in the NF period, the trend declines. In the F period (mature state of ozone recovery,

see Fig. 1S), the AoA trends are approaching the magnitudes of the Ref period. In the EMAC-L90 REF-C2 simulation the AoA trend shows the smallest magnitude in the Ref period and an increase in the NF and F periods. For a detailed intermodel comparison of the AoA changes refer to Eichinger et al. (2018).

### 3.2 Reasons for the minimal AoA trend in the extratropical stratosphere

In the first part of this section we analyze the vertical shift of pressure levels and connect it with the net upward shift of the
circulation (Shepherd and McLandress, 2011; Singh and O'Gorman, 2012; Oberländer-Hayn et al., 2016). In the second part we estimate the effect of the correction for the vertical shift of the pressure levels on AoA, AbM and RCTT trends and analyze the processes leading to the minimal AoA trend in the Ex regions. This part of the analysis is based solely on the CMAM simulation. However, the findings of Eichinger et al. (2018) show that the AoA distribution and its change are governed by similar processes among the different CCMI-1 REFC2 models. This suggests that our results can be considered
robust also for other CCMI-1 simulations.

### 3.2.1 Vertical shift and stratospheric shrinkage

Observations and models have shown that the tropopause shifts upward (Santer et al., 2003; Añel et al., 2006) together with the whole tropospheric circulation pattern (Singh and O'Gorman, 2012) due to the tropospheric warming and stratospheric cooling in the course of climate change. The tropospheric warming also influences BDC wave driving by causing an upward
displacement of the critical layers for wave breaking (Shepherd and McLandress, 2011). The effect of the upward shift of the circulation on the BDC trends have been highlighted recently by Oberländer-Hayn et al. (2016), where the shift has been divided between the shift of pressure levels and relative to pressure levels. Our methodology (see Section 2.2) is developed to diagnose (and subtract) the vertical shift of pressure levels, which is diagnosed as a trend of geopotential height of pressure levels (Fig. 3).



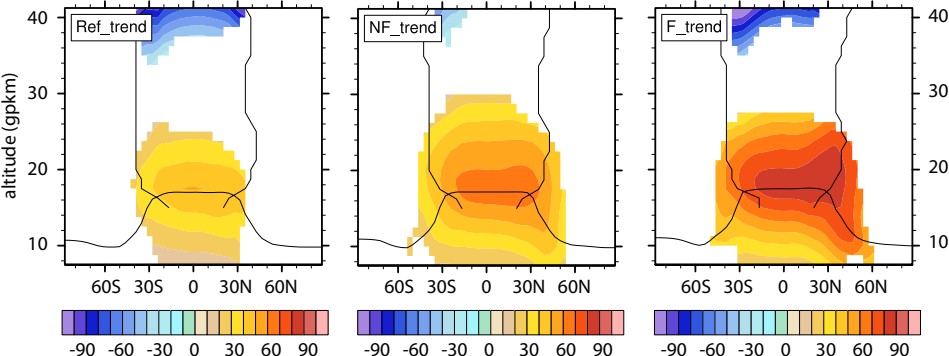


**Figure 3.** CMAM trend of geopotential height of pressure levels [gpm/decade] interpolated to the climatological geopotential height of the selected pressure levels in each period. The mean tropopause and turnaround latitude positions in the DJF season are marked with black lines. Only the trends in the regions where they exceed the statistical significance of 95% confidence level are plotted.


In Fig. 3 we see that the tropopause is collocated with the region of the largest upward shift trend of pressure levels in all periods. The trend of the pressure levels in the vicinity of the tropopause in the tropics is around 40 *m/decade*, 60 *m/decade* and 80 *m/decade* in the Ref, NF and F period, respectively. Above the tropopause, the trend is decreasing with altitude and it is not significant higher up. Starting in the middle stratosphere (above around 38 gpkm), the trend is negative. This vertical

structure of the pressure level geopotential height trend characterizes the stratospheric shrinkage.

The vertical shift is not globally homogenous, it shows a maximum between approximately 30°S and 30°N in the Ref period and it is not significant in the polar regions. In the NF and F period, the area of significance of the trend is widening, moving upwards and shifting slightly to the NH. The magnitude of the upward shift trend is smallest in the Ref period and largest in the F period. Interestingly, there is a strong (around 200 *m/decade*) almost significant (at a 85% confidence level and so not

shown) negative trend of the pressure levels in the Ref period in the SH polar stratosphere between 10 to 25 gpkm connected most likely with radiative effects due to ozone depletion and dynamical response of the SH polar vortex.

As described in eq. (2), the height changes of the pressure levels result in a dependence of the AoA trend on the vertical coordinate system. Since the partial derivative of AoA with respect to the geopotential height is generally positive in the stratosphere (see Fig. 2), the sign of the second term in eq. (2) is determined by the local derivative of the geopotential height

of pressure levels (in our approximation by the trend of geopotential height of pressure levels). Hence, the AoA trend in geopotential height vertical coordinate is smaller/larger than in pressure coordinates, where the pressure levels rise/sink. In the LS, the pressure levels rise and so the AoA trend is smaller (more negative) in geopotential height than in pressure



coordinates. This can be easily illustrated by assuming a situation, where the AoA trend in pressure coordinates would be zero. But as the pressure levels rise, the fixed geopotential corresponds to increasing pressures over time. These are

connected with smaller AoA, which yields a negative AoA trend in geopotential coordinates.

The second part of the vertical shift of the circulation, the vertical shift relative to the pressure levels, is diagnosed in the literature mainly with relation to the tropopause rise. E.g., Abalos et al. (2017) assessed the shift by accounting for the tropopause rise by means of remapping to tropopause relative coordinates. The tropopause rise in geopotential height vertical coordinate and relative to the surrounding pressure levels is shown in Fig. 4.

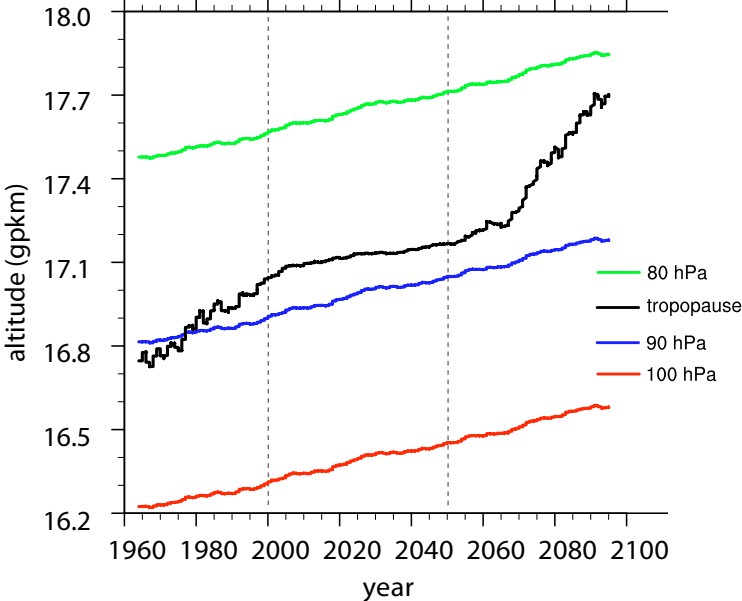


**Figure 4.** Time evolution of CMAM zonal mean geopotential height [gpkm] of selected pressure levels and of a lapse rate first tropopause averaged between 30°S and 30°N smoothed by a decadal running average.

The tropopause trend in the geopotential height vertical coordinate can be used in our methodology (Section 2.2, eq. 3)

instead of the trend of geopotential height of pressure levels to diagnose the trends also in the tropopause relative coordinate. However, due to the neglection of the variable vertical shift of pressure levels (see Fig. 3; stratospheric shrinkage (Lübken et al., 2009)), the assumption of a uniform shift equal to the tropopause rise everywhere in the stratosphere may lead to an increasing overestimation of the upward shift effect with distance from the tropopause.




In next section, we estimate the effect of the vertical shift of pressure levels on the computation of trends. Additional impact
of the vertical shift relative to the pressure levels (included in the tropopause rise) on the trend computation is quantified for
the trend of the net upwelling only, as the upwelling will undoubtedly reflect the tropopause rise (Oberländer-Hayn et al.,
2016), but this is not as certain for the regions higher in the stratosphere.

### 3.2.2 The effect of the upward shift and AoA isoline widening

In Fig. 5 we show the AoA, AbM, RCTT trends in the Ref, NF and F period computed from spatial averages over the Ex
regions (between 20 gpkm and 25 gpkm and 20°- 50°N/S) in the standard geopotential height coordinates and after the
correction for the vertical shift of the pressure levels (denoted by the superscript $^c$). The bars in Fig. 5 show the trends of the
net upwelling proxy computed directly by integration of the mass flux proxy between the turn-around latitudes ($UP_{wstar}$, see
Section 2.3) at 20 gpkm before (blue) and after the correction (red) to the vertical shift of pressure levels and also after the
correction to the tropopause shift (green). The vertical level of 20 gpkm has been chosen, because it corresponds with the
lower boundary of the Ex regions. In Fig. 5 we have chosen to lower the significance threshold to the 90% confidence level,
because many trends are significant between the 90% and 95% levels. The information from Fig. 5 is summarized in Tabs.
1S and 2S in the Supplement, where also values of trends significant at least at the 80% confidence level are given.

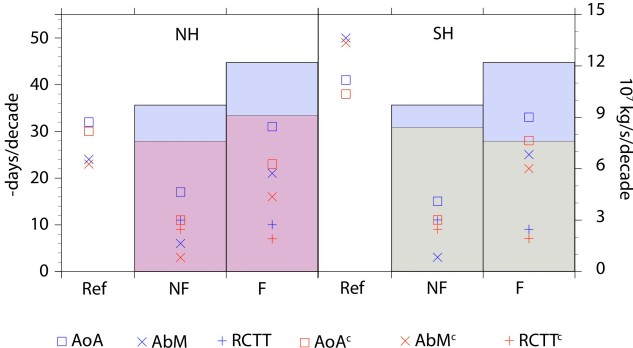

**Figure 5.** Trends of AoA (squares), AbM (crosses) and RCTTs (pluses) averaged over the Ex regions (ExNH on the left,
ExSH on the right) in the Ref, NF and F period in - *days/decade*. Blue markers denote the trends in geopotential height
vertical coordinates and Red markers the trends computed after the correction for the vertical shift of pressure levels. Bars
represent $UP_{wstar}$ trends at 20 gpkm before (blue) and after the correction (red) for the vertical shift of pressure levels and
also after the correction for the tropopause shift (green) given in $10^7 kgs^{-1} / decade$ . AbM and RCTT trends are computed
in a reduced Ref period (1970-2000). Presented are only trend values significant at least at the 90% confidence level.




In Fig. 5, both before and after the correction for the pressure level shift, we can see the strongest negative AoA and AbM trends in both Ex regions in the Ref period. The AoA and AbM trends both have the same time evolution, decrease (in absolute values) from Ref to NF and increase again in the F period. On the contrary, both before and after the correction, the RCTT trends are not significant in the Ref period (significant at the 80% confidence level before the correction in ExNH - see Tab. 1S). In the NF period, the RCTT trends are slightly larger than in the F period both before and after the correction. The AoA trends in the Ex regions are dominated by RCTT trends in the NF period. The time evolution of AoA, AbM and RCTT trends is unchanged after the correction for the pressure level shift.

The difference between the directly computed trend value and the trend value after the correction gives us an estimate of the influence of the pressure level shift on the trend computation. The influence is determined by the second term on the right side of eqs. (1) or (2), which consists of the vertical gradient of the given quantity and the rate of the pressure level shift that is identical for all quantities, but differs between periods. We have seen in Fig. 5 that for the AoA and AbM trends, the influence grows in future periods in accordance with the time evolution of the trend of geopotential height of pressure levels (Fig. 3). Absolute values of the AoA trend are reduced by 2, 6 and 8 *days/decade* after the correction in ExNH and by 3, 4, 5 *days/decade* in the ExSH for the Ref, NF and F period respectively. For the AbM trend, the reduction is smaller (maximally 5 *days/decade* in ExNH in the F period), but in the ExSH region in NF, the AbM$^c$ trend is no longer significant. RCTT trends are also reduced in absolute value by 2 and 3 *days/decade* in the NF and F period in ExNH and by 2 *days/decade* in both future periods in the ExSH region.

In the Ref period, the UPwstar trends geopotential height coordinates (see blue bars in Fig. 5) are not significant (only at the 80% confidence level shown in Tab. 2S). After the correction to the pressure level shift (red bars), the UP$_{wstar}$ trends are reduced by 22% and 25% in the NF and F period respectively. In NF and F, both before and after the correction, the UP$_{wstar}$ trend is larger in the F than NF period (as it is the case also for AoA and AbM trends).

It has been noted e.g. by Randel et al. (2008), Butchart et al. (2011) and Butchart (2014) that the residual circulation changes mainly depend on strengthening of the tropical upwelling. We find a good correspondence with the RCTT and RCTT$^c$ trends only for the UP$_{wstar}$ trends corrected for the tropopause shift (Fig. 5, green bars). The correction for the tropopause rise is implemented in a similar manner as for the pressure level shift (eq. (3)); only the tropopause trend instead of the variable pressure level trend is cumulatively subtracted. The UP$^{rel}_{wstar}$ trend is larger in the NF than F period agreeing with the time evolution of RCTT trends in the Ex regions. The UP$^{rel}_{wstar}$ trend is larger than UP$^c_{wstar}$ trend in NF because, only in the NF period, the tropopause rises less rapidly than the surrounding pressure levels in the CMAM simulation (see Fig. 4). These CMAM results agree well with the results of Oberländer-Hayn et al. (2016) who showed that there is no increase of upwelling when accounting for the tropopause rise. We also did not find any net tropical upwelling trends (differs from UP by containing the time evolving density) after the correction to the pressure level shift and only one positive trend in the NF period after the correction for the tropopause shift (see Tab. 3S in the Supplement). In summary, we cannot unambiguously link the time evolution of AoA, AbM and RCTT trends in the Ex regions with the tropical upwelling trends.



We have also analyzed trends of spatial averages of RC in the Ex regions. RC (i.e. the residual circulation strength) is a local
measure of acceleration of the residual circulation defined in Section 2.3. However, the RC trends are only sparsely
significant at the 80% confidence level and severely reduced in magnitude (see Tab. 1 in the next section or Tab. 1S in the
Supplement) after the correction for the vertical shift of pressure levels. The possible link between the time evolution of
AoA, AbM and RCTT trends and the acceleration of the residual circulation together with a possible role of wave driving is
analyzed on seasonal basis in Section 3.3.


With the methodology for correction to the vertical shift of pressure levels (eq. 3), we can now demonstrate the effect of
vertical shift on the occurrence of minimal AoA trends in the Ex regions. The distribution of AoA$^c$ trends is shown in Fig. 6.
Without the effect of vertical shift of pressure levels (upward shift up to about 30 gpkm; Fig. 3), the AoA trends do not form
the pronounced, localized, almost symmetric minima in the Ex regions as in Fig. 2 (upper plots for CMAM). Only the global
minimum in the Ref period remains in the ExSH region. This is likely due to the impact of ozone depletion in the SH in the
Ref period and the methodological limitation that we do not account for the pressure levels shift in the SH midlatitudes and
polar region due to the lower significance of the vertical shift trend than our threshold (95% confidence level, Fig. 3).

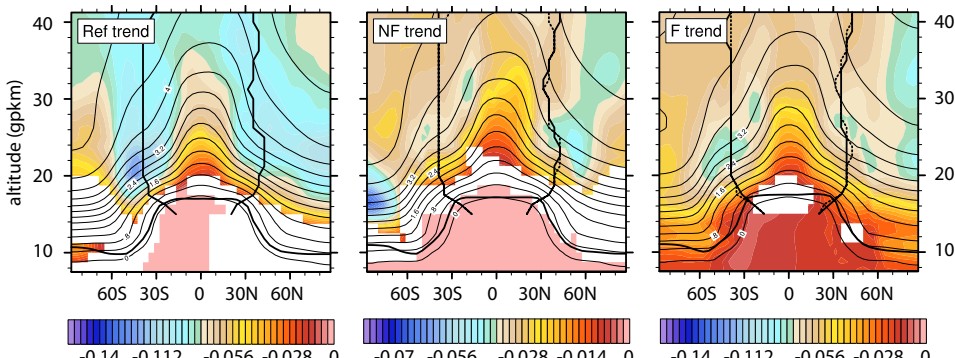

**Figure 6.** CMAM zonal mean AoA annual trend [year/decade] after the correction of the pressure levels trend for the Ref
(left), NF (middle) and F (right) period, in colors. Only trends that exceed the 95% significance level are plotted. The mean
AoA climatology of the respective period is contoured and the tropopause as well as the turnaround latitudes (computed
from the residual circulation field interpolated to the geopotential height corrected to the upward shift of the pressure levels)
are shown by bold black lines. The turn-around latitude of the preceding period is illustrated by a dashed line.

Fig. 6 also shows the mean turn-around latitude positions. The mean turn-around latitude positions between the periods show
only small differences in their meridional location. In the NH, there are some visible changes towards a narrowing of the





upwelling region below about 26 gpkm and towards a widening above. This is in agreement with the results of Hardimann et al. (2014), who found that the tropical upwelling region narrows below about 20 hPa, and widens above. This pattern also appears for turn-around latitude position changes in seasons (not shown), when changes in the SH are pronounced as well.

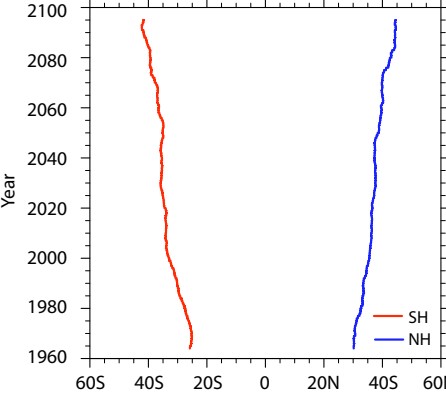


**Figure 7.** Time evolution of the meridional position of the zonal mean AoA=2years isoline at 20gpkm after the correction to the upward shift of pressure levels for CMAM. The evolution is smoothed by a decadal running mean.

Fig. 7 shows that the AoA distribution in the Ex regions is widening also after subtraction of the vertical shift of pressure
levels. This widening is related only to AoA isolines and is completely independent on any circulation widening. In Fig. 7 we show the time evolution of a meridional position of intersection of the AoA = 2 years isoline with a lower boundary of the Ex regions (20 gpkm) after the correction to the vertical shift. The AoA = 2 years isoline crosses the Ex regions initially on their equatorward flanks and the intersection with 20 gpkm moves poleward from 40°S and 40N by the end of the period. This is a consequence of the AoA distribution and negative AoA$^c$ trend (Fig. 5).

Moreover, in the Ex regions, there is a specific role of AbM for the AoA isoline widening. The climatological distributions of RCTTs (left panel) and AbM (right panel) in the period 1970-2000 overlaid by the AoA climatological distribution (contours) are shown in Fig. 8. The basic features of the CMAM climatologies in Fig. 8 are similar to the RCTT and AbM climatologies from the CCMI-1 REF-C1 simulations (see Figs. 1 and 2 in Dietmüller et al., 2018). Note that the RCTT distribution up to around 30 gpkm between the turn-around latitudes is much broader than the AoA distribution and the
horizontal gradient of the RCTTs is smaller. The difference between AoA and RCTTs (i.e. AbM) is biggest in the Ex regions and slightly above (up to about 28 gpkm). In this region of the climatological AbM maximum, the AoA distribution is narrower and has a sharper horizontal gradient than the RCTTs. Due to the negative trend in AbM, the AoA distribution further widens as it becomes more similar to the RCTT distribution. Besides the negative AoA$^c$ trend and decreasing effect



of AbM, the vertical shift relative to pressure levels (tropopause shift) can also be contributing to the widening shown in Fig.

7 in the sense of shifting the climatological AoA isolines upward.

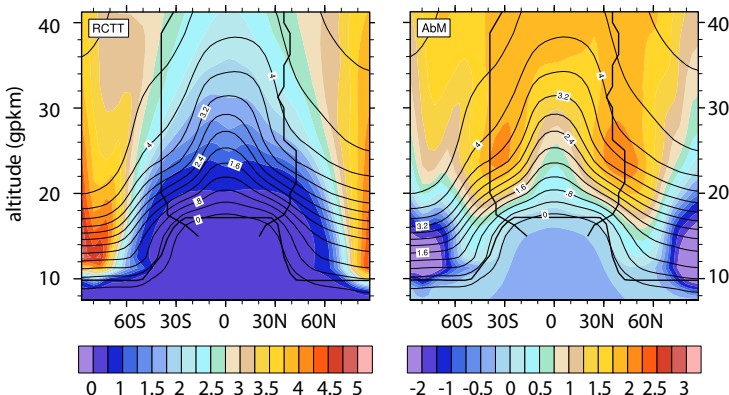

**Figure 8.** CMAM zonal mean RCTTs [year] on the left and aging by mixing on the right averaged over the 1980-2000

period. The thick black lines show the mean tropopause and the turnaround latitude positions in the Ref period.

### 3.3 Acceleration of the residual circulation and wave driving

The Ex regions lie in the upper flank of the shallow BDC branch. There, AoA has been found to be controlled by both horizontal and vertical residual circulation tendencies and the horizontal mixing tendency (Ploeger et al., 2015b, see their Fig. B1). From Fig. 5 we know that the AoA trends in the Ex regions are dominated by the AbM trends in the Ref and F and

by the RCTT trends in the NF period. Note that AbM is not a local quantity. Ploeger et al. (2015b; Lagrangian model study) found that AbM corresponds well with the local mixing integrated along the air parcel pathway. Garny et al. (2014) and Ploeger et al. (2015a) found that the AbM trends are affected predominantly by changes of the local mixing intensity in the lowest part of the stratosphere, but those further above are strongly coupled to the residual mean circulation. This complicates the analysis of possible AbM drivers in the Ex regions, because the Ex regions are located in the transition

region between these two regimes. RCTTs are an integrated quantity with climatological values around one year in the Ex regions (Fig. 8) and hence they can be influenced by changes in seasonal UP and RC strength (this holds for AbM as well). To gain a better insight into these connections, we here study how the seasonal RC, UP, local wave drag and vertically integrated drag trends correlate with each other and with the time evolution of AbM and RCTT trends in the Ex regions.





The Ex regions are located in a region characterized by domination of the meridional residual mean velocity component $\overline{v}^*$

(see the aspect ratio analysis of Birner and Bönisch (2011)). There is a local direct relationship between $\overline{v}^*$ and the collocated zonal mean force per unit mass ($\bar{F}$) derivable from the Transformed Eulerian Mean equations (Andrews et al., 1987) under the quasi-geostrophic scaling:

$$\overline{v}^*(\varphi, z) = -\frac{\bar{F}(\varphi, z)}{f},$$
(4)

where $\varphi$ is the latitude and $f$ the Coriolis parameter. Eq. 4 provides a direct link between the strength of residual circulation
in the extratropical stratosphere and the distribution of the zonal mean zonal force. Besides the trends of the spatially averaged local zonal mean zonal force, we compute also upwelling trends by evaluating the quasi-geostrophic version of the DC integral (Haynes et al., 1991) at the turn-around latitudes (UP$_{DC}$, see Section 2.3). UP$_{DC}$ provides information on wave driving in the whole vertical domain of our analysis. The total drag is in our case composed by the resolved wave driving characterized by EPFD and OGWD and NOGWD from parameterizations. In the Ex regions NOGWD is by an order of
magnitude smaller than OGWD. Thus OGWD largely controls the net GWD that is analyzed further. As explained before in Section 2.3, in the definition of forces we also replace zonal mean density by zonal mean density averaged over a period to filter out the effect of negative density trends.

In Tab. 1 we provide annual and seasonal trends of the spatial averages of total drag (TD), RC, EPFD and GWD scaled by the climatological zonal mean density and corrected for the vertical shift of pressure levels ($^c$). As the choice of the
significance threshold is a subjective process, we are providing in Tabs. 1 and 2 also values of trends significant between the 80% and 90% confidence level, which are denoted with a superscript $^+$. As explained in the Appendix, EPFD trends are computed from spatial averages between 18 gpkm and 25 gpkm and 15°- 30°N/S and GWD is averaged in the poleward part of the Ex regions between 18 gpkm and 25 gpkm and 30°- 45°N/S near its climatological minimum. The values of trends including the density are shown in Tab. 5S in the Supplement.

Tab. 1 shows that neither TD$^c$ in the ExNH nor its components over the smaller regions have significant annual trends. Only in the ExSH, GWD$^c$ has a significant annual trend in all periods and EPFD$^c$ and TD$^c$ in the NF and F period. This is partly reflected by the RC$^c$ annual trend at a weaker significance level ($^+$) in the ExSH region. Seasonally, RC$^c$ and TD$^c$ and its components have significant trends in both Ex regions. The seasonal trends have the largest magnitude in absolute value in both Ex regions in the Ref period in the DJF and MAM season. In JJA and SON the trends are smaller in absolute value in
the ExNH or reverse sign towards weakening of the drag (except GWD) and deceleration of the circulation in the ExSH. In the remainder of this section, we will focus on seasonal trends in the NF and F periods and on the issue of different evolution of trend (RCTT$^c$ vs. AbM$^c$) in the future periods.


**Table 1.** Upward shift corrected trend values ($^c$) of total drag (TD, EPFD + GWD) and its components EPFD and GWD

multiplied by $\langle \overline{\rho} \rangle_{period}$ (all in $10^{-5} kg m^{-2} s^{-2} / decade$) and RC ($10^{-3} kg m^{-2} s^{-1} / decade$) averaged over the Ex

regions in the Ref, NF and F period. Presented are trend values significant at least at the 90% confidence level or at least at

the 80% confidence level (denoted by superscript $^+$), otherwise the cell is left blank.

| Period | Ref | | | | | NF | | | | | F | | | | |
|---|---|---|---|---|---|---|---|---|---|---|---|---|---|---|---|
| Season | DJF | MAM | JJA | SON | *Ann* | DJF | MAM | JJA | SON | *Ann* | DJF | MAM | JJA | SON | *Ann* |
| | | | | | | | | *ExNH* | | | | | | | |
| TD$^c$ | -3.5 | -1.4$^+$ | -0.5 | -1.2 | | | -1.4 | -1.1 | -0.5$^+$ | | | -1.4 | | -0.7$^+$ | |
| EPFD$^c$ | -1.5 | -0.6$^+$ | | -0.4$^+$ | | | -0.8 | -0.7 | -0.3$^+$ | | -0.8 | -0.7 | -0.4$^+$ | | |
| GWD$^c$ | -3 | -1.5 | | | | | -1.3 | -0.4 | | | | -1.2$^+$ | | | |
| RC$^c$ | 3.8 | | 1 | 2.3 | *1.7$^+$* | | 1.1 | 0.8 | 1.2 | | 1.8$^+$ | 2 | | 2.6 | |
| | | | | | | | | *ExSH* | | | | | | | |
| TD$^c$ | -2.2 | -1.1 | 1.4 | 2.3 | | -1.9 | | | -1.1$^+$ | *-0.9* | -1.2 | -0.7$^+$ | | | *-0.8* |
| EPFD$^c$ | | -0.8 | 0.9 | | | -1 | | | | *-0.5* | -0.6 | -0.5$^+$ | -0.8$^+$ | | *-0.6* |
| GWD$^c$ | -0.5 | -0.4$^+$ | -1.2 | | *-0.6* | -0.6 | -0.5$^+$ | | -1.8 | *-0.7* | -0.3$^+$ | | | -0.9 | *-0.4* |
| RC$^c$ | 3.2 | 1.4 | | -3.7 | | 1.2 | 1.6 | 1.1 | | *0.1$^+$* | 1.1 | 1.4 | 2.2 | | *0.1$^+$* |

First, note that the seasonal TD$^c$, EPFD$^c$ and GWD$^c$ trends are generally larger in the NF than F period in both Ex regions,

which excludes the possibility of a direct link between the AbM trends and local wave drag. For RC$^c$, we can see stronger

trends in F than NF in the MAM and SON season in the ExNH region and in JJA season in the ExSH region. In other

seasons the trend is not significant in both future periods or is higher in the NF period. In ExNH, all drag trends are

accompanying RC$^c$ trends in MAM for both future periods and in JJA only for the NF period. However, the drag trends in

MAM do not reflect almost doubling of the magnitude of the MAM RC$^c$ trend between the NF and F period. In the ExSH

region, local drag trends accompany the RC$^c$ trend in both future periods only in DJF. In this season, the absolute value of

trends decreases more strongly for drag than for RC$^c$ between NF and F. In the ExSH, the link with RC$^c$ is especially poor

for the GWD$^c$ trends. The reason for this may be that in the ExSH region, GWD$^c$ is only weakly correlated with $\overline{v}^*$ (annual

mean correlation equal to 0.31, see Tab. 4S).

In Tab. 2 we show seasonal UP trends computed with different methodologies after the correction for the vertical shift of

pressure levels. The trends of net upwelling (including time evolving density) are given in Tab. 6S in the Supplement. From

the three methodologies, two based on the residual mean velocities (UP$^c_{wstar}$, UP$^c_{vstar}$) yield almost identical results regarding

seasonal occurrence of the trends. The time evolution of the seasonal UP$^c_{wstar}$ and UP$^c_{vstar}$ trends varies between seasons. In

the DJF season both trends are largest in Ref and smallest in the F period, but in MAM the trends are largest in F and

smallest in the NF period. This is true also for the annual trends and trends in other seasons that are not significant in the Ref

period.


**Table 2.** Trends of UP (in $10^7 kgs^{-1}/decade$) computed with three different methods after the correction for the vertical shift of pressure levels ($UP^c_{wstar}$, $UP^c_{vstar}$ and $UP^c_{DC}$) in the Ref, NF and F period. Presented are trend values significant at least at the 90% confidence level or at least at the 80% confidence level (denoted by superscript $^+$), otherwise the cell is left blank.

| | | | | | | | | | | | | | | | |
|---|---|---|---|---|---|---|---|---|---|---|---|---|---|---|---|
| | | | | | | | **Upwelling** | | | | | | | | |
| **Period** | **Ref** | | | | | **NF** | | | | | **F** | | | | |
| **Season** | **DJF** | **MAM** | **JJA** | **SON** | *Ann* | **DJF** | **MAM** | **JJA** | **SON** | *Ann* | **DJF** | **MAM** | **JJA** | **SON** | *Ann* |
| **$UP^c_{DC}$** | | | | | | | $5.2^+$ | $4.8^+$ | | | | | 6.9 | 7.3 | |
| **$UP^c_{vstar}$** | 22 | 6.4 | | | | 11 | 5 | $3.7^+$ | | $5.3^+$ | 8.3 | 9.2 | $5.5^+$ | 6.6+ | $6.7^+$ |
| **$UP^c_{wstar}$** | 25.7 | 8.8 | | | | 15.8 | $5.9^+$ | $5.3^+$ | $7.1^+$ | 7.6 | 13 | 11.3 | 8.6 | | 9.1 |


The $UP^c_{DC}$ inferred from the integral of $TD^c$ at the turn-around latitudes have only weakly significant seasonal trends in MAM and JJA in the NF period and significant trends in the JJA and SON season in the F period. The JJA trend is higher in F than NF. We can only speculate that the $UP^c_{DC}$ trends in MAM and JJA in the NF period can be connected with the local drag trends in the ExNH (Tab. 1), but this is almost certainly not possible for the $UP^c_{DC}$ JJA and SON trends in the F. One possible reason is that the seasonal $UP^c_{DC}$ trends in the F period are driven by changes of drag above the Ex regions, where $TD^c$ has been spatially averaged. The $RC^c$ trends correlate well with the seasonal $UP^c_{wstar}$ and $UP^c_{vstar}$ trends, with the exception of the strong $UP^c_{wstar}$ and $UP^c_{vstar}$ trends in DJF in the NF period accompanied by small (missing in the ExNH) $RC^c$ trends in this season and period.

In summary, after the correction for the vertical shift of pressure levels, the $RCTT^c$ trends are stronger in NF than in F in both Ex regions. $AbM^c$ trends are weak or not significant in NF and dominate the AoA trend in the F period (Fig. 5). In the ExSH region, when significant, the seasonal $RC^c$ trends are only slightly higher in F than in NF. Nevertheless, in ExNH, the seasonal $RC^c$ trends are more than twice as large in F than in NF. This correlates better with the $AbM^c$ trend evolution in the ExNH. However, in the ExSH region, we cannot link the magnitude of the seasonal $RC^c$ trends to the time evolution of the $AbM^c$ trend. The $AbM^c$ trend in ExSH is not significant in NF and then larger in F than the ExNH $AbM^c$ trend. Seasonal trends of $TD^c$, $EPFD^c$, $GWD^c$ show some small increase from the NF period to the F period in ExNH, but rather the opposite in the ExSH region. This does not allow a clear link to the time evolution of the $RC^c$ or $AbM^c$ and $RCTT^c$ trends. Trends of another drag based quantity ($UP^c_{DC}$) show a clear strengthening of the integrated $TD^c$ in the F period, but are only sparsely significant with different seasonality than the upwelling estimates based on the residual mean circulation. The $UP^c_{wstar}$ and 500    $UP^c_{vstar}$ trends also do not allow a clear link to the time evolution of $AbM^c$ or $RCTT^c$ trends, as they have a different time evolution in the DJF and MAM and JJA seasons between the periods.

To conclude, after the subtraction of the vertical shift of pressure levels the $AbM^c$ and $RCTT^c$ trends in the Ex regions cannot be easily linked to the upwelling or local residual circulation changes. The link to the wave driving trends in the NF and F



period is even less clear. This indicates that additional mechanisms may be involved, for example the effect of stratospheric
shrinkage could explain some of the AoA changes. Moreover, our diagnostic methods, in particular regarding the sparse
spatiotemporal sampling of GWD effects, may not meet the needs for accurate analysis of the connections between the
processes.

## 4 Discussion

### 4.1 Speeding up of the BDC and wave driving

In several studies a physical connection between changes in AoA, AbM and RCTTs, the speeding up of the residual mean
circulation (see Birner and Bönisch, 2011; Li et al., 2012; Garny et al., 2014; Ploeger et al., 2015a) and increasing wave
driving by changes in the resolved and unresolved extratropical wave forcing (e.g. Okamoto et al., 2011; Shepherd and
McLandress, 2011; Butchart, 2014) has been postulated. In the present study, after the correction to the vertical shift of
pressure levels, we could not find a simple link between trends of wave driving ($TD^c$, $GWD^c$, $EPFD^c$ and $UP^c_{DC}$) and net
upwelling ($UP^c_{wstar}$ and $UP^c_{vstar}$, Fig. 5 and Tab. 2) and local residual circulation strength ($RC^c$) in the extratropical lower to
middle stratosphere (Tabs. 1 and 2). Also, a clear connection to the trends of $AoA^c$, $AbM^c$ and $RCTT^c$ (Fig. 5), i.e. quantities
representing transport connected with the BDC, is missing. The fact that our results do not fully support the hypothesis
indicates that additional mechanisms may be influencing transport and dynamics in the studied regions (as discussed in 4.2),
but it also partly stems from peculiarities of our analysis. Namely, our analysis is focused on the Ex regions, which do not
belong fully to the altitudinal range of the shallow BDC branch and lie e.g. on the interface between the two dynamical
regimes influencing the AbM (Garny et al., 2014; Ploeger et al., 2015a). Also, our approach of dividing the REFC2
simulations into three periods is different from the above mentioned studies and brings along the novel opportunity to
analyze the time evolution of trends of different quantities on the expense of more difficult acquisition of significance.

The trends computed in geopotential height vertical coordinate after the correction for the pressure level shift ($^c$) should be
comparable to the trends in pressure levels (eqs. (1) and (2)). Hence, it is exactly due to the methodology (choice of
significance levels, periods, seasons, trends vs. differences) that we did not find as robust changes of wave drag in
connection with the acceleration of the residual mean circulation as reported in other studies based on pressure coordinates
(e.g. Okamoto et al., 2011; McLandress and Shepherd (2009), Shepherd and McLandress, 2011). However, this does not
hold for analyses based on log-pressure coordinates, as they include the effect of the vertical shift of pressure levels (see
discussion in 4.2). If the vertical shift of pressure levels is not subtracted, the emerging link between the trends or changes in
general may be simply induced by the common impact of the vertical shift and not by the structural changes.

We would like to point out one particular result of our wave driving analysis - a week correspondence between $GWD^c$ and
the $RC^c$ trends. Although the quasi-geostrophic theory itself does not support GWs, it is often used to study GW effects on
the circulation. As shown in Appendix (Fig. A1), the correlation between $\overline{v}^*$ and GWD is weaker than for EPFD, suggesting




that the quasi-geostrophic local relationship (eq. 4) is less valid for GWD. Also the weak correspondence between GWD and
RC trends argues against a direct relationship (of absolute values) as in eq. (4). We do not argue that the lower validity of the
local relationship (eq. 4) for GWD implies that the quasi-geostrophic approximation is generally unfit for studies of the
GWD effects. We argue that the complex role of GWD on transport in the stratosphere cannot be judged from the zonal
mean monthly mean data output only, as the average value is not fully representative of the GWD distribution.

GWs are intermittent (e.g. Hertzog et al., 2012; Wright et al., 2013) and asymmetrically distributed (Hoffmann et al., 2013;
Šácha et al., 2015; Hoffmann et al., 2016; Pišoft et al., 2018) in nature. This intermittency and asymmetry of the spatial
distribution of GWD (OGWD in particular) is to some extent present also in the CCMI-1 simulations. For example, a crucial
role of the zonally asymmetric OGWD distribution for its interannual variability has been shown by Šácha et al. (2018) for a
CMAM specified dynamics simulation (McLandress et al., 2013), which uses the same parameterization of orographic GWs
as the CMAM REFC2 simulation analyzed in this study. The zonal mean data can hide different effects on the residual
circulation due to the zonally asymmetric distribution of the GWD (Šácha et al., 2016) present in the models and monthly
mean output can mask the extreme GWD values. Note that also the widely used DC principle relies on zonally symmetric
forces (Haynes et al., 1991). Clearly, there is a need for provision of as frequent as possible 3D GWD output (complex, not
only the induced zonal acceleration component) as possible in connection with reporting of extreme values during the time
window in addition to average values to properly diagnose the possible GW effects present in the models.

**4.2 Stratospheric shrinkage**

In Section 3.2.1 (Fig. 3) we have analyzed the vertical shift of pressure levels, which results in a so-called stratospheric
shrinkage. Although our methodology accounts for the effect of the vertical shift of pressure levels (stratospheric shrinkage)
in the process of trend computation, the effect of decreasing geometrical distances between pressure levels in the course of
the model simulation, which can directly influence the AoA, RCTTs and AbM, cannot be quantified in our analysis. To our
knowledge, this effect has not been mentioned yet in relation to the possible causative factors of the AoA trend (or BDC
acceleration) before. For example in Tab. 3 we show how the mean distance between the 1 hPa and 100 hPa levels will
change between the 1960s and the 2090s. In the analyzed simulations, the 100 hPa level in the tropics will be closer to the 1
hPa level by about 400-700 meters in the 2090s than in the 1960s. Depending on the variable geopotential height of pressure
levels and distances between the two levels at the start of the analysis in the simulations, the differences in Tab. 3 are ranging
from 2.33 % for CMAM to 1.3% for NIWA of the original distance in the 1960s in the tropics. Assuming a constant speed of
advection in the vertical, this directly reduces the RCTTs (for vertical velocity of an order of $10^{-4}\,ms^{-1}$ it takes 84 days less
to travel the distance shrunken by 720 meters). The effect on AbM or mixing in general can be of more complex, possibly
nonlinear, nature.






**Table 3.** The change in the mean distance (in geopotential meters) between the 1hPa and 100hPa levels in the tropics between the 1960s and the 2090s for the analyzed REFC2 CCMI-1 simulations.

| CMAM | EMACL47r2 | EMACL90 | HadGEM3 | GEOS | NIWA_ens |
|---|---|---|---|---|---|
| 721 ± 268 | 622 ± 230 | 658 ± 281 | 682 ± 186 | 646 ± 250 | 409 ± 285 |

As pointed out before, we also cannot account directly for the shift relative to the pressure levels. Typically in the literature
(Oberländer-Hayn et al. (2016), Abalos et al. (2017)), the tropopause is taken as a proxy for the upward shift relative to pressure levels. Otherwise, the possibly non-homogeneous shift relative to pressure levels in the stratosphere cannot be objectively assessed. Note also that the shift related to the tropopause can become very complicated to disentangle. The whole region changes its structure because of the increasing occurrence of double and multiple tropopauses due to increasing baroclinicity in the course of climate change (Castanheira et al., 2009; Castanheira and Gimeno, 2011; Wang and Polvani,
2011; Añel et al. 2012).

 In Fig. 4 (Section 3.2.1) we have shown how the first lapse rate tropopause in CMAM shifts relative to pressure levels between 30°S and 30N for the CMAM REFC2 simulation. The averaged tropopause shifts by about 1 gpkm in the course of the simulation. Such a shift is in the range of the tropopause trends from the Coupled Model Intercomparison Project (CMIP5) ensemble mean in the tropics (Fig. 7a in Vallis et al. (2015)). It corresponds also to the net equatorial tropopause
height change from the simulations analyzed by Oberländer-Hayn et al. (2016) and Abalos et al. (2017), who diagnosed a net tropopause shift of 430 m in global average. In the whole period of the analysis, the tropopause pressure changes by almost 10 hPa. Depending on the vertical shift of the stratopause (assuming that it will follow the pressure levels rather than the tropopause), the tropopause shift causes additional stratospheric shrinkage.

Fig. 4 also shows that the rate of the tropopause shift relative to pressure levels almost perfectly follows the division into the
periods used in this study. The tropopause rises rapidly relative to pressure levels in the Ref and F period, when we found the biggest AbM and AoA trends (Fig. 5). In the NF period, there is no visible shift of the tropopause relative to pressure levels, which corresponds well with the small (ExNH) or insignificant (ExSH) AbM trends. The rate of the tropopause shift (and possibly of the net stratospheric shrinkage) thus correlates with the time evolution of AbM trends. However, at this stage we cannot provide a detailed or analytical description of the mechanism.

It has been shown before by Shepherd and McLandress (2011) that the EPFD changes associated with GHG increases in the subtropics are largely controlled by the upward displacement of the critical layers for wave breaking, and by McLandress and Shepherd (2009) and Okamoto et al. (2011) that the OGWD changes are linked to the upward shift of the subtropical jet (Son et al., 2009). Also, Eichinger et al. (2018) found that the mixing changes, as well as the inter-model spread are connected to changes (upward shift) of the background PV gradient in the CCMI-1 simulations. All of those studies were
based on pressure coordinates and so the shift they are referring to is the shift relative to pressure levels (tropopause shift).





The RCTT trends do not reflect the time evolution of the shift relative to pressure levels directly (i.e. the trend is larger in the NF than in the F period). The reason can lie in the strong dependence of RCTTs on the tropical upwelling. Tropical upwelling is the only quantity (as explained in Section 3.2.1) for which we computed the trends also in coordinates corrected to the tropopause shift (Tab. 2S, 3S and 6S in the Supplement). After the correction to the tropopause shift, the net tropical

upwelling shows the same time evolution of trends as RCTTs. This is caused by a missing vertical shift of the tropopause relative to pressure levels in NF in CMAM. Future research is needed regarding a possible cause and robustness of this feature between the models.

Finally, there are important consequences for trend analyses based in log-pressure coordinates in connection to the stratospheric shrinkage. Unlike in the pressure coordinates, the vertical shift of pressure levels is reinstated in log-pressure

coordinates due to the utilization of a constant scale height ($H$) in the conversion from the pressure coordinates. Therefore the trends computed in log-pressure coordinates are influenced by the effect of the variable vertical shift of pressure levels. Moreover, the choice of constant $H$ also leads to the artificial increase of the log-pressure vertical velocity (see the Supplement of Dietmüller et al., 2018 for details on the transformation) due to neglecting the shrinking geometrical distance between the pressure levels.

**5 Summary and conclusion**

In a subset of CCMI-1 REFC2 simulations, we have pointed out a remarkable pattern of similarity in the morphology of the stratospheric AoA trend, especially in the future periods (2000-2050 and 2050-2100). These are the regions of minimal AoA trends, which are located in both hemispheres between 20 gpkm and 25 gpkm and 20°- 50°N (ExNH) and 20°- 50°S (ExSH). We hypothesized that these minima are connected with the climatological AoA gradient and the previously known AoA

trend drivers: a) upward shift of the circulation (Oberländer-Hayn et al., 2016; Abalos et al., 2017) and b) decreasing AoA trend consisting of the RCTT (residual circulation transit times) and AbM (aging by mixing) contribution (Garny et al., 2014). Both mechanisms also influence the widening of the AoA isolines in the Ex regions.

From the net upward shift of the circulation, the part connected with the vertical shift of the pressure levels has been diagnosed and the so far neglected stratospheric shrinkage pattern has been pointed out. We then showed that the AoA,

AbM, RCTT (in the Ex regions) and the net tropical upwelling trends are reduced when accounting for the vertical shift of pressure levels. Moreover, the local residual circulation strength (RC) does not exhibit any trends when accounting for the shift. After the vertical shift subtraction, we have found only seasonal trends of the total zonal mean wave drag and its components (resolved and unresolved) and we could not find a direct relationship between them and the seasonal RC or upwelling trends. Also, in the future periods, we could not find any clear link between the time evolution of the AbM and

RCTT trends and wave driving, RC or upwelling trends after the correction for the shift of pressure levels. This indicates that additional mechanisms may be involved. For example, we discuss a mechanism how the stratospheric shrinkage can affect the AoA changes. Moreover, our diagnostic methods, in particular regarding the sparse spatiotemporal sampling of GWD



effects, may not meet the needs for accurate analysis of the connections between the processes in the models, for which more detailed GWD output would be needed.


The analysis is based on geopotential height coordinates, but the argument that the upward shift (together with AoA isoline widening) is necessary for the visual pattern of localized AoA trend minima in the extratropical stratosphere holds also in pressure coordinates. The location of the minimal AoA trends is in the best vertical range for the AirCore measuring tool (Engel et al., 2017) and their easy visual detectability makes them the best regions for AoA trend observations. The detection

of the localised trend minima in the Ex regions in observations could provide validation of the processes that lead to their formation in the models. Those are the upward shift of the circulation, the AoA decreasing trend and most importantly its aging by mixing (AbM) component that can be connected with the fine dynamical features of the model's lower stratosphere. To gain more insight in future climate projections, we particularly suggest inter-model analysis of the stratospheric shrinkage, including the time evolution of the vertical shift of the tropopause, and its effect on the stratospheric

circulation.

***Data availability.*** All data CCMI-1 used in this study can be obtained through the British Atmospheric Data Centre (BADC) archive (ftp://ftp.ceda.ac.uk, last access: August 2018). For instructions for access to the archive see http://blogs.reading.ac.uk/ccmi/ badc-data-access.).

**Appendix: Issue of compensation and specification of the averaging domain for wave drag components**

There are indications that changes in the unresolved wave drag are often compensated by changes in the resolved wave driving (Mclandress and McFarlane, 1993; Cohen et al., 2013; Cohen et al., 2014). This so called "compensation mechanism" is present also in comprehensive climate model projections of the BDC change (Sigmond and Shepherd, 2014) and complicates the possibility to clearly separate the effects of individual wave drag components. The compensation needs

to be taken into account to identify the areas, where the individual drag components can influence the advection to the Ex regions. Therefore, we analyze the wave drag distribution and the occurrence of compensation near the Ex regions.
In Fig. A1a, we show the total drag (GWD+EPFD) climatology (contours) overlaid over the ratio of unresolved to resolved wave drag (GWD/EPFD). Figs. A1b and c show the climatological distribution of EPFD and GWD, respectively. We see that GWD has its climatological minimum slightly below the Ex regions around 18 gpkm and is dominant (bigger than

EPFD) in the lower stratosphere only in the NH (Fig. A1a, red color). This extratropical lower stratospheric region is located exactly around the NH turn-around latitude from 16 to 22 gpkm. In the location of the GWD minimum in SH, GWD is smaller than EPFD. Other regions of GWD dominance are scattered higher in the stratosphere.
 In the extratropical stratosphere, regions with the same sign of GWD and EPFD prevail (Fig. A1a, red and antique white colors). The ratio is shown for the Ref period and differs only slightly in the NF and F period when corrected to the pressure



levels shift (not shown). The regions of the lower stratospheric minima of GWD are collocated with the "saddle-like regions"
in the EPFD distribution around 16 to 22 gpkm (Fig. A1b). Those regions are positioned on the upper flank of the
subtropical jets of both hemispheres, but are more pronounced in NH, where the GWD is stronger. The total drag distribution
(contours in Fig. A1a) largely copies the EPFD distribution, but it is smoother in the lower stratosphere as the "saddle-like"
pattern from the EPFD distribution is filled by GWD. This can be considered as a fingerprint of the compensation and

indeed, this region includes the 70 hPa level, where Cohen et al. (2013) demonstrated the compensating effects between
OGWD and EPFD and NOGWD for driving of the residual circulation.

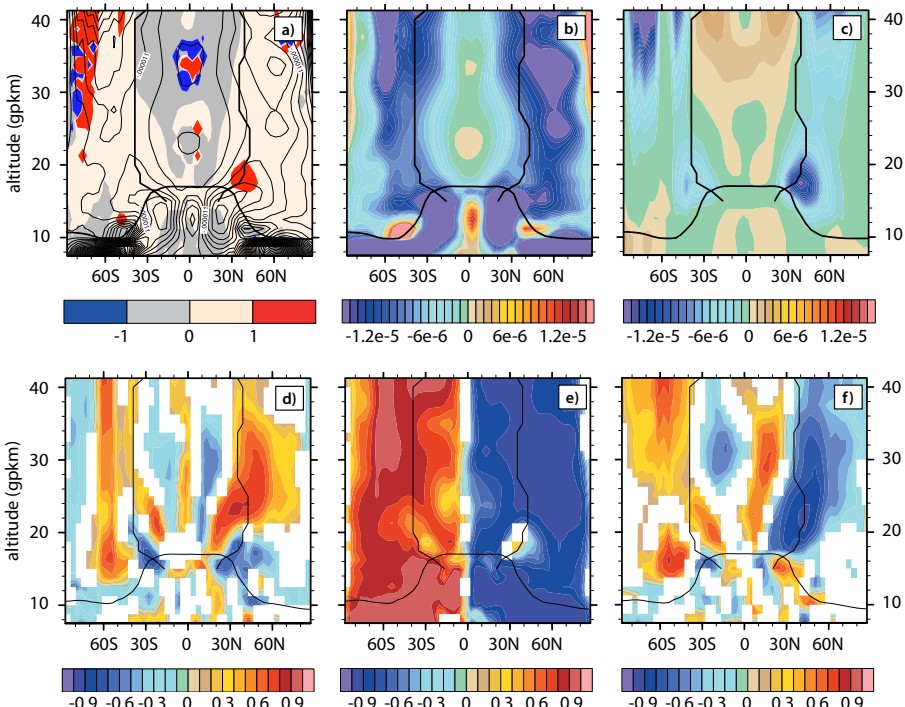

**Figure A1.** (a) GWD/EPFD ratio of the Ref period. The color scale is chosen so that the regions with the absolute value of

zonal mean zonal GWD stroger than EPFD are highlighted. The contours show the total drag Ref climatology. (b) EPFD and

(c) GWD climatology of the Ref period (units $m \cdot s^{-2}$). Correlations between (d) GWD and EPFD, (e) EPFD and $\overline{v}^{*}$ and (f)

GWD and $\overline{v}^{*}$ in the Ref period. The vertical axis is in geopotential kilometers [gpkm]. The plots are overlaid by a mean



tropopause and turnaround latitude positions in the Ref period. Only regions with a statistical significance of correlations exceeding the 95% confidence level are plotted.


Further information on the compensation is provided in Figs. A1d, e, f. Here we show correlations between the drag components and $\overline{v}^{*}$. For brevity, only the annual mean correlations in the Ref period are shown, although the compensation between wave drag components and the relationship with the residual circulation is changing during the year (see Tab. 4S in the Supplement for the correlations computed after the correction to the vertical shift of pressure levels).

In Fig. A1d, the distribution of negative correlations, which indicate compensation between the drag components, agrees well with the location of regions of minimum (strongest) GWD (EPFD saddle regions). Otherwise, in the extratropical stratosphere we find mainly positive GWD and EPFD correlations. The localization of the compensation is confirmed also by correlations of GWD and EPFD with $\overline{v}^{*}$ (Fig. A1e, f). In the extratropical stratosphere, EPFD does not have significant correlation with $\overline{v}^{*}$ solely in the region of minimal GWD in the NH (low value of correlation also in the location of the SH

GWD minimum). GWD is strongly correlated with $\overline{v}^{*}$ in those regions. Elsewhere, the high absolute values of correlation between EPFD and $\overline{v}^{*}$ (generally higher than 0.7) show a good validity of the direct relationship between the zonal mean zonal force and $\overline{v}^{*}$ (eq. 4), although neither EPFD nor $\overline{v}^{*}$ are computed using a quasi-geostrophic formula. The correlation of GWD with $\overline{v}^{*}$ has a more patchy structure and does not reach as high correlation values as for EPFD. In ExNH, GWD is better correlated with $\overline{v}^{*}$ than in the ExSH region.

Due to the climatological distibution and ocurrence of compensation, we do not average the drag components over the whole Ex regions. Instead, EPFD trends are computed from a spatial average between 18 gpkm and 25 gpkm and 15°- 30°N/S, i.e. slightly lower and more equatorward than the location of Ex regions. There, EPFD dominates the driving of the advection to the Ex regions. GWD is averaged in the poleward part of the Ex regions between 18 gpkm and 25 gpkm and 30°- 45°N/S near its climatological minimum. For direct comparison with the RC trends, the total drag is averaged over the whole Ex

regions. The occurrence of compensation within the averaging domain means that the total drag can have a significant trend, regardless of significance of the trends of its components. Because the compensation is also present around the turn-around latitudes at 20 gpkm, where we evaluate the DC integral for computation of the mean upwelling (UP$_{DC}$), we do not separate the individual drag components (UP$_{DC}$ trends are given in Tab. 4).

***Competing interests.*** The authors declare that they have no conflict of interest.

***Special issue statement.*** This article is part of the special issue "Chemistry–Climate Modelling Initiative (CCMI) (ACP/AMT/ESSD/GMD inter-journal SI)". It is not associated with a conference.





***Author contributions.*** PŠ performed all the analyses and wrote the article together with RE. JA, LdT, PP and HG made substantial contributions to the conception of the study and interpretation of the results and participated in drafting the article. SD provided the RCTTs data, helped with their analysis and commented on the paper. In their role as CCMI model PIs, the other authors contributed information concerning the analyzed models, commented on the manuscript and helped to revise the paper.


***Acknowledgements.*** We acknowledge the modelling groups for making their simulations available for this analysis, and the joint WCRP SPARC/IGAC Chemistry-Climate Model Initiative (CCMI) for organizing and coordinating this model data analysis activity. We thank the British Atmospheric Data Center (BADC) for hosting the CCMI-1 data archive. We acknowledge the UK Met Office for use of the MetUM. The EMAC simulations have been performed at the

German Climate Computing Centre (DKRZ) through support from the Bundesministerium für Bildung und Forschung (BMBF). DKRZ and its scientific steering committee are gratefully acknowledged for providing the HPC and data archiving resources for the consortium project ESCiMo (Earth System Chemistry integrated Modelling). The NIWA programme CACV was supported by the NZ Government's Strategic Science Investment Fund (SSIF). The authors wish to acknowledge the contribution of NeSI high-performance computing facilities to the results of this research. New Zealand's national

facilities are provided by the New Zealand eScience Infrastructure (NeSI) and funded jointly by NeSI's collaborator institutions and through the Ministry of Business, Innovation & Employment's Research Infrastructure programme (https://www.nesi.org.nz). The GEOSCCM is supported by the NASA MAP program and the high-performance computing resources were provided by the NASA Center for Climate Simulation (NCCS).

The study was supported by the Government of Spain under grant no. CGL2015-71575-P and Petr Šácha was also partly

supported by GA CR under grant nos. 16-01562J and 18-01625S. Further, Petr Šácha would like to acknowledge Felix Ploeger and Mohamadou Diallo for their comments and suggestions at the beginning stages of the analysis and would like to thank Rachel White and Joan Alexander for their comments and discussions about the wave driving and compensation mechanism.

Roland Eichinger, Hella Garny and Simone Dietmüller acknowledge funding from the Helmholtz Association under grant

VH-NG-1014 (Helmholtz-Hochschul-Nachwuchs-forschergruppe MACClim). Olaf Morgenstern acknowledges funding by the New Zealand Royal Society Marsden Fund (grant 12-NIW-006). N. Butchart and the development of HadGEM2-ES was supported by the joint DECC/Defra Met Office Hadley Centre Climate Programme (GA01101) and the European Commission's 7th Framework Programme StratoClim project 226520. Petr Pišoft was supported by GA CR under grant nos. 16-01562J and 18-01625S. Juan A. Añel was partially supported by a 'Ramón y Cajal' Fellowship (RYC-2013-14560).





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
