# Peer review of "Extratropical Age of Air trends and causative factors in climate projection simulations"

_Atmospheric Chemistry and Physics, 2018_

## Referee Comment (RC1) · Anonymous Referee #2 · 9 Feb 2019

This paper analyzes trends in age of air, the stratospheric circulation, mixing and wave driving in CCM output. A novel aspect of the analysis is the inclusion of an adjustment for the change in geopotential height of pressure surfaces. This adjustment explains one of the main features in the age of air trends. The analysis is well done and helps further our understanding of model age of air trends in the recent past and future. My main criticism is on the section describing the individual wave forcing trends that is too detailed and would be better off mostly moved to the supplement. I suggest publication with consideration of the specific comments below.

Specific comments:

Line 16: 'in the course' should be rephrased. Maybe 'associated with' or something similar.

Lines 17-18: You say that the 'mechanisms' of BDC strengthening are 'well understood' but also that there are 'open questions concerning its dynamical driving'. This seems contradictory so it should be rephrased. Maybe by 'mechanisms' you mean something different, like 'features'.

Line 18: 'accessing' should be 'assessing'

Line 21: you should define 'gpkm' the first time you use it

Line 23: remove 'a' before 'climate change'. Remove 'also' after 'But'.

Line 25: add comma after 'Furthermore'.

Line 49: '..which makes it difficult to clearly...'

Line 54: 'accessing' should be 'assessing'

Line 55: remove 'an' before 'age-of-air'. Also I think you meant 'stratosphere' and not 'troposphere'

Line 62: 'In the following sections, where we...'

Line 65: '...parameterization schemes and has been used in previous studies...'

Lines 66-67: remove 'that have been based on CMAM'

Line 71: remove 'see the scheme in'

Line 74: change 'us' to 'an'

Line 91: change 'with' to 'by'

Line 163: add 'a' before 'daily basis'

Line 172: spell out 'especially'

Line 179: 'we state several times...'

Line 186: add 'the' before 'form'

Line 209: 'extratropical' is misspelled. Also remove 'the scheme in'.

Figure 2: Colorbar tick marks are so thick it's hard to see the colors.

Lines 240-245: Really hard to see the changes in trends between the periods due to the different scaling as you mention. You might think about how to adjust this if you really want the readers to see the differences between the time periods.

Line 273: 'the trend decreases with altitude…'

Line 277: 'the trend widens, moving…'

Line 301: 'due to neglecting the variable…'

Line 304: add 'the' before 'next'

Lines 311-314: The description of the trends and how they're shown in Figure 5 is a bit confusing. It took me a while to figure out that you're showing the same value in the blue bars in both the NH and SH sides of the plot, and that the red and green bars don't have to do with NH or SH but only with the different corrections. I would suggest thinking about a different way of showing this. Probably best to have a third plot labeled 'Tropics' that just shows the bars, blue, green and red so it's clear that you're comparing the tropical upwelling flux trends. Then the NH and SH plots could be narrowed quite a bit since they will just have the symbols.

Line 364: add 'a' before 'seasonal'

Figure 6 (and 2): Since you show trends in days/decade in Figure 5 it would be nice to see the other trend figures in those units. Also helpful to label the colorbars in the figures.

Line 390: change 'on' to 'of'

Line 393: change 'from' to 'of'

Line 422: remove 'here'

[Figure]

Line 433: change 'by' to 'of'

Line 435: add comma after 'Thus'

Section 3.3: This section is tough to get through without getting bogged down in trying to understand all of the different trends shown in Tables 1 and 2 and what they mean. I would highly suggest trying to distill this section and put the tables in the supplement. You really get the main points across in the first and last two paragraphs of the section.

Line 532: change 'week' to 'weak'

---

## Referee Comment (RC2) · Anonymous Referee #1 · 4 Mar 2019

This paper compares trends in the stratospheric Brewer-Dobson circulation from 5 different CCMI-1 models in terms of mean age, residual circulation, aging by mixing and wave drag trends. The study focuses on the subtropical lower stratosphere regions where all models considered robustly show largest negative mean age trends. A new and interesting aspect of the study is the consideration of the effect of the vertical shift of pressure levels under climate change on the age trend. The authors argue that the effect of such a shift should be largest in the subtropical lower stratosphere where the mean age gradient is largest, and that the shrinkage of the stratosphere likely contributes to the strong negative trends in that region.

The paper is overall well written and the results are well presented. The topic is of high relevance for the atmospheric community and the effect of a vertical shift has not

been sufficiently discussed so far. I therefore recommend publication, but have a few comments to be addressed below.

Minor comments:

1. I find it difficult to follow the discussion of wave driving in Sect. 3.3. Most arguments here are based on numerous numbers in several tables. Condensing the key information into a figure would be very helplful for the reader, e.g. to illustrate that AbM and RCTT can not easily be linked to the local quantities.

2. This point is related to the one above and concerns the missing clear link between wave drag changes and the residual circulation acceleration (e.g., stated on P1, L26ff). When looking at Fig. A1, on the contrary, there seems to be a strong (anti-) correlation between the meridional resid. circulation velocity and EPFD in the upper part of the Ex region and with GWD in the lower part of the Ex region - as expected. I guess that this is due to the different time scales considered: The stated missing link concerns long-term trends, the correlation in Fig. A1 is dominated by shorter term variability. Nevertheless, I find Fig. A1 very interesting and would suggest to move it to the main part, or a zoomed in version of it showing just time series in the Ex region and their (non-) correlation. It would then be very interesting to investigate further at which time scale the correlation between residual circulation and wave drag changes breaks down (e.g., by filtering out specific parts of the variability), and maybe similarly for correlations between other quantities (related to my comment 1).

3. I find the argumentation that the AoA distribution widens due to AbM and RCTT changes a bit hand-wavy (e.g., P16, L402). Maybe including global trends of AbM and RCTT in Fig. 8 could be useful for making the point clearer?

Specific comments:

P1, L17: What is the distinction here between "mechanisms for BDC strengthening" and "dynamical driving"? The former is said to be well understood, the latter still open

- please clarify.

P1, L26: Why are residual circulation, upwelling and wave driving here presented as individual drivers? Actually they should be all closely connected.

P4, L96ff: I would find a table containing the main information regarding the different models considered helpful.

P4, L116: I don't get the point why RCTTs and AbM starts 1970. Largest RCTTs are about 5 years. Thus for the simulations starting in 1960 it should be possible to have RCTTs already in 1965. Is it because a longer spin-up is needed for having reliable AoA and AbM?

P5, L141: Shouldn't the relation Y=Y' be just the general property of a scalar function?

P6, L172 (and throughout the paper): A somewhat picky note regarding the notation of TEM quantities: Usually the star is placed next to the overbar and not below.

P14, L336ff: I would prefer presenting percentage changes here (as is the case for the upwelling changes in the next paragraph).

P22, L535: The authors state a "weak correspondence between GWD and RC trends", however Fig. A1 shows a strong correlation between about 18-21 km. I guess this point is again related to the time scales considered (see comment 2 above). Please clarify.

Technical comments:

P2, L48: McLandress

P2, L53: Maybe better to say "best available/commonly used tools"?

P2, L62ff: Complicated sentence. Maybe change like: 2Here, we produce ... by using ...

P3, L65: has been chosen

P3, L65: Maybe better: "studies regarding BDC"

P3, L66: Maybe better: climate change based on CMAM

P3, L74: shows the additional

P8, L211: does slightly

P8, L214: Delete "text"

P11, L271: "upward shift trend" - I would suggest to write either "shift" or "trend".

P13, L304: In the next...

P14, L343: trends in geopotential

P16, L389: Figure 7...

P16, L393: moves polewards to

P18, L429: Equation 4 provides...

P18, L445: Table 1 shows...

P23, L584: Figure 4...

P25, L646: Mc Landress

P26, L670: stronger

Fig. 7: In the NH the x-label 40 has an "S" instead of "N".

---

## Author Comment (AC1) · 27 Apr 2019

Response to Anonymous Referee #2:
(our response in italics)

This paper analyzes trends in age of air, the stratospheric circulation, mixing and wave driving in CCM output. A novel aspect of the analysis is the inclusion of an adjustment for the change in geopotential height of pressure surfaces. This adjustment explains one of the main features in the age of air trends. The analysis is well done and helps further our understanding of model age of air trends in the recent past and future. My main criticism is on the section describing the individual wave forcing trends that is too detailed and would be better off mostly moved to the supplement. I suggest publication with consideration of the specific comments below.

*We are grateful for the reviewer's positive review, valuable comments and suggestions, which helped us to improve the quality of the paper.*
*We have adapted all of your suggestions, please refer to the revised, marked up version of the manuscript.*
*Following is our reply to the comments that led to more substantial changes than rewording:*

- Figure 2: Colorbar tick marks are so thick it's hard to see the colors.
- Lines 240-245: Really hard to see the changes in trends between the periods due to the different scaling as you mention. You might think about how to adjust this if you really want the readers to see the differences between the time periods.

*Figure 2 has been modified to a uniform colour scale and the problem with tick marks has been fixed.*

- Lines 311-314: The description of the trends and how they're shown in Figure 5 is a bit confusing. It took me a while to figure out that you're showing the same value in the blue bars in both the NH and SH sides of the plot, and that the red and green bars don't have to do with NH or SH but only with the different corrections. I would suggest thinking about a different way of showing this. Probably best to have a third plot labeled 'Tropics' that just shows the bars, blue, green and red so it's clear that you're comparing the tropical upwelling flux trends. Then the NH and SH plots could be narrowed quite a bit since they will just have the symbols.

*Figure 5 has been modified according to your comments. Third plot "Tropics" has been added, plots narrowed and the symbols are thicker in the revised version.*

- Figure 6 (and 2): Since you show trends in days/decade in Figure 5 it would be nice to see the other trend figures in those units. Also helpful to label the colorbars in the figures.

*Figure 6 and 2 has been modified accordingly to the units days/decade.*

- Section 3.3: This section is tough to get through without getting bogged down in trying to understand all of the different trends shown in Tables 1 and 2 and what they mean. I would highly suggest trying to distill this section and put the tables in the supplement. You really get the main points across in the first and last two paragraphs of the section.

*Thank you very much for your suggestion. Section 3.3 has been shortened, Tables 1 and 2 have been moved to the Appendix (Tabs. A1, A2) together with the description of the seasonality of the trends.*
*New table (Tab. 1) has been created showing only annual local residual circulation and wave driving trends. One paragraph has been rephrased and one added to summarize that neither the annual nor seasonal local trends can be unambiguously linked to the AbM and RCTT trends.*
*See also our response to the Referee #1 regarding the wave driving section.*

---

## Author Comment (AC2) · 27 Apr 2019

**Response to Anonymous Referee #1:**
(our response in italics)

This paper compares trends in the stratospheric Brewer-Dobson circulation from 5 different CCMI-1 models in terms of mean age, residual circulation, aging by mixing and wave drag trends. The study focuses on the subtropical lower stratosphere regions where all models considered robustly show largest negative mean age trends. A new and interesting aspect of the study is the consideration of the effect of the vertical shift of pressure levels under climate change on the age trend. The authors argue that the effect of such a shift should be largest in the subtropical lower stratosphere where the mean age gradient is largest, and that the shrinkage of the stratosphere likely contributes to the strong negative trends in that region.

The paper is overall well written and the results are well presented. The topic is of high relevance for the atmospheric community and the effect of a vertical shift has not been sufficiently discussed so far. I therefore recommend publication, but have a few comments to be addressed below.

*We are grateful for the reviewer's positive review, valuable comments and suggestions, which helped us to improve the quality of the paper. Especially we acknowledge the comments on wave driving and the role of variability in the relationship between wave drag and residual circulation.*
*We have adapted almost all of your suggestions, please refer to the revised, marked up version of the manuscript. Following is our reply to your minor and specific comments:*

Minor comments and specific comment P22L535:

1.I find it difficult to follow the discussion of wave driving in Sect. 3.3. Most arguments here are based on numerous numbers in several tables. Condensing the key information into a figure would be very helplful for the reader, e.g. to illustrate that AbM and RCTT can not easily be linked to the local quantities.

*Based also on the comments from Ref#2, we have shortened the Sect. 3.3. Tables 1 and 2 have been moved to the Appendix (Tabs. A1, A2) as well as the text concerning the seasonality of the trends.*
*Instead of a figure that Ref#1 suggests, new table (Tab. 1) has been created showing only local residual circulation and wave driving trends on annual basis. One paragraph has been rephrased and one added (P19L453ff) to summarize that neither the annual nor seasonal local trends can be unambiguously linked to the AbM and RCTT trends.*

2. This point is related to the one above and concerns the missing clear link between wave drag changes and the residual circulation acceleration (e.g., stated on P1, L26ff). When looking at Fig. A1, on the contrary, there seems to be a strong (anti-) correlation between the meridional resid. circulation velocity and EPFD in the upper part of the Ex region and with GWD in the lower part of the Ex region - as expected. I guess that this is due to the different time scales

considered: The stated missing link concerns long-term trends, the correlation in Fig. A1 is dominated by shorter term variability. Nevertheless, I find Fig. A1 very interesting and would suggest to move it to the main part, or a zoomed in version of it showing just time series in the Ex region and their (non-) correlation. It would then be very interesting to investigate further at which time scale the correlation between residual circulation and wave drag changes breaks down (e.g., by filtering out specific parts of the variability), and maybe similarly for correlations between other quantities (related to my comment 1).

P22, L535: The authors state a "weak correspondence between GWD and RC trends", however Fig. A1 shows a strong correlation between about 18-21 km. I guess this point is again related to the time scales considered (see comment 2 above). Please clarify.

*We would like to thank very much the Ref#1 for this minor and specific comment. We followed his/her recommendation and examined how the correlation evolves when filtering out parts of the variability. This is done by smoothing the time series (monthly mean data from 1960 to 2100) by a running average with increasing number of months included in the average. As an illustration, we show in Fig. D1 the evolution of a correlation between GWD and $\overline{v}^*$ in the part of the ExNH region after the correction for a vertical shift of pressure levels. Indeed, the correlation gets smaller in absolute value when filtering out larger parts of the variability.*

[Figure]

**Figure D1.** *Evolution of a correlation between GWD and $\overline{v}^*$ in the part of the ExNH region after filtering out parts of the variability and after the correction for a vertical shift of pressure levels. On the x axis is the number of months included in the running average, on the y axis is a value of the correlation.*

*For EPFD we observe similar, but smaller decrease - including 240 months in the running average gains a correlation of about -0.7. What is interesting, for the correlation between $\overline{v}^*$ and the total drag on a broader domain (whole ExNH region) the decrease is negligible and for 240 months in the running average we have a correlation of -0.95. This indicates that the compensation mechanism (or*

*better the whole complex of compensation mechanisms) on a broader domain allows using the quasi-geostrophic scaling (i.e. neglecting the terms with zonal mean zonal wind that would otherwise enter the eq. 4 in the manuscript) with almost perfect precision. This is obviously not true for the individual drag components and especially for GWD.*

*As it became clear during the discussion that the wave driving section should be shortened, we decided not to include those new results to the manuscript. The Section 3.3 is now focused to give a clear message that there is no unambiguous link between the local quantities and AbM and RCTT trends.*
*Nevertheless, in our future research (as also pointed out in the Discussion) we intend to produce a detailed study of the role of GWD for the BDC and transport in the models in general.*

3. I find the argumentation that the AoA distribution widens due to AbM and RCTT changes a bit hand-wavy (e.g., P16, L402). Maybe including global trends of AbM and RCTT in Fig. 8 could be useful for making the point clearer?

*The argumentation concerning widening has been shortened and reworded (P16L396ff) to list all mechanisms leading to the AoA isoline widening and to highlight the specific role of the decreasing AbM. The AbM trends are not shown in the paper, but a reference to the study Eichinger et al. (2019) has been added (P16L402), where in their Fig. 3 (blueish colors where dominant) the distribution of the effect of the AbM change on the AoA change is shown.*

Other specific comments that led to more substantial changes than rewording:

P4, L96ff: I would find a table containing the main information regarding the different models considered helpful.

*On P4L100 we have added two references to the papers, where information on the simulations used in our study is summarized in tables.*

P4, L116: I don't get the point why RCTTs and AbM starts 1970. Largest RCTTs are about 5 years. Thus for the simulations starting in 1960 it should be possible to have RCTTs already in 1965. Is it because a longer spin-up is needed for having reliable AoA and AbM?

*As RCTTs are calculated as backward trajectories, it is unclear at the start how many years the trajectories will need to reach the tropopause. To be on the safe side, we generally use a buffer of 10 years for that. In hindsight, you are right, most trajectories need less than 5 years, but still, not all of them. So to be correct, it should probably be a 6 years buffer, which would give us at the maximum 4 extra years of RCTTs. We are confident that our results are robust enough with the (even) 30 years used in the Ref period and that these 4 extra years will not have a considerable impact on them.*

P5, L141: Shouldn't the relation Y=Y' be just the general property of a scalar function?

*We think that strictly mathematically speaking Y and Y' are not identical, because they are functions of different variables.*

P6, L172 (and throughout the paper): A somewhat picky note regarding the notation of TEM quantities: Usually the star is placed next to the overbar and not below.

*Thank you for noticing this. The notation of TEM quantities has been changed through the main text and Appendix.*

P14, L336ff: I would prefer presenting percentage changes here (as is the case for the upwelling changes in the next paragraph).

*As suggested, we have added information on percentage changes (P14L339ff in the revised manuscript).*